# ^68^Ga-HBED-CC-WL-12 PET in Diagnosing and Differentiating Pancreatic Cancers in Murine Models

**DOI:** 10.3390/ph16010080

**Published:** 2023-01-05

**Authors:** Qiying Xiang, Danni Li, Chao Cheng, Kai Xu, Changjing Zuo

**Affiliations:** 1School of Medical Imaging, Xuzhou Medical University, Xuzhou 221004, China; 2Department of Nuclear Medicine, The First Affiliated Hospital (Changhai Hospital) of Naval Medical University, Shanghai 200433, China

**Keywords:** PD-L1, pancreatic cancer, PET-CT, ^68^Ga, immune microenviroment

## Abstract

Positron emission tomography (PET) has been proven as an important technology to detect the expression of programmed death ligand 1 (PD-L1) non-invasively and in real time. As a PD-L1 inhibitor, small peptide WL12 has shown great potential in serving as a targeting molecule to guide PD-L1 blockade therapy in clinic. In this study, WL12 was modified with HBED-CC to label ^68^Ga in a modified procedure, and the biologic properties were evaluated in vitro and in vivo. ^68^Ga-HBED-CC-WL12 showed good stability in saline and can specifically target PD-L1-positive cells U87MG and PANC02. In PANC02-bearing mice, ^68^Ga-HBED-CC-WL12 showed fast permeation in subcutaneous tumors within 20 min (SUV_max_ 0.37) and was of higher uptake in 90 min (SUV_max_ 0.38). When compared with 18F-FDG, ^68^Ga-FAPI-04, and ^68^Ga-RGD, ^68^Ga-HBED-CC-WL12 also demonstrated great image quality and advantages in evaluating immune microenvironment. This study modified the ^68^Ga-labeling procedure of WL12 and obtained better biologic properties and further manifested the clinical potential of ^68^Ga-HBED-CC-WL12 for PET imaging and guiding for immunotherapy.

## 1. Introduction

The presence of immunotherapy is a breakthrough following traditional therapies. In a healthy immune system, immunoactivation and immumosuppression are in a dynamic balance. In many cancer types, T-cell activation is inhibited by the programmed death receptor 1 (PD-1), which can interact with programmed death ligand 1 (PD-L1), over-expressed on cancer cells, leading to escape of these cancers from immune surveillance [1,2]. In addition, the high expression of PD-L1 tends to be related to poor prognosis and high mortality [3]. Biologicals targeting the PD-1/PD-L1 axis have been widely researched, and some antibodies have been approved by Food and Drug Administration (FDA) for certain cancer treatments, such as anti-PD-L1 antibody nivolumab [4], pembrolizumab [5], and anti-PD-L1 antibody atezolizumab avelumab. However, a series of cancers demonstrate low response rates of 15–25% [6]. Research has shown that PD-L1 expression is closely related with PD-1/PD-L1 blockade response rate. Therefore, to better exert the efficacy of PD-1/PD-L1 blockade, it is necessary to observe the dynamic PD-L1 status and take it as a marker for predicting response to therapy [7]. 

Currently, immunohistochemistry (IHC) staining of tumor biopsy is used to evaluate the PD-L1 status, but local and invasive biopsies cannot provide adequate information due to the high heterogeneity within and between patients, and single time-point biopsies cannot monitor dynamic changes of PD-L1 in treatment regimens [8,9]. Molecular imaging technology such as nuclear medicine can provide quantitative, real-time, and non-invasive assessment of target expression that enable therapeutic monitoring [10]. 

The potential of radiolabeled anti-PD-L1 antibodies, nanobodies, and small inhibitors to image PD-L1 expression have been demonstrated in different tumor models [11,12,13,14,15]. Atezolizumab [16], the first FDA-approved PD-L1 mAbs, has been utilized for PET imaging. ^89^Zr-atezolizumab predicts the therapeutic effect of atezolizumab in patients by assessing the PD-L1 status [17]. However, the inherent characteristics of mAbs limit its further exploration as radiotracers. The mAbs have two major defects, one of which is their slow clearance rate due to the poor tumor penetration and long metabolism time in vivo, leading to a long metabolism of 4 to 7 days to obtain immune images with acceptable target-to-background ratio [18]. The other one is that therapeutic antibodies would competitively bind to PD-L1 target to interfere with the accurate assessment of PD-L1 by antibody-based molecular imaging. Therefore, the research hotspot turned to the non-mAb-based tracers with better biological properties. New probes with shorter biological half-live periods and faster pharmacokinetics can provide patients with real-time information and produce high-contrast images to guide treatment.

One of PD-L1-targeted peptides, WL12, which is a cyclic peptide comprising 14 amino acids, binds to PD-L1 with high affinity (IC50 ≈ 23 nM). The analogs labeled with ^64^Cu and ^68^Ga were developed and have shown potential to detect graded levels of PD-L1 expression in vivo in preclinical models of several cancer types [14,15]. ^68^Ga-NOTA-WL12 has been evaluated clinically in non-small cell lung cancer patients and has demonstrated its safety and feasibility [19]. In this study, we evaluated the modified WL12 with HBED-CC as the coupling group to modify the labeling procedure with moderate conditions and obtain better biologic properties. The stability and specific affinity in pancreatic tumors were evaluated preclinically to explore the suitable time point for image acquisition. In the meantime, PET imaging of some radiotracers targeting other bioactive sites or biochemical procedures were compared to that of ^68^Ga-HBED-CC-WL12 (abbreviated as ^68^Ga-WL12 below).

## 2. Results

### 2.1. ^68^Ga-Labeling and Stability In Vitro 

HBED-CC-WL12 was of a moderate labeling conditions with a radiochemical purity of higher than 90% in the pH range of 3–6, which extended the applied pH value in preparation of DOTA- or NOTA-based radiopharmaceuticals (Figure 1). ^68^Ga-HBED-CC-WL12 was less stable in 10% FBS and was of 56.5 ± 2.3% RCP after 60 min incubation. ^68^Ga-HBED-CC-WL12 was stable in saline with 92.5 ± 1.5% RCP after 60 min incubation, meeting the requirement for temporary storage. 

### 2.2. Cellular Uptake 

To evaluate the binding capacity of ^68^Ga-WL12 to their target protein PD-L1, a cellular uptake assay was performed on PANC02 and U87MG cells. As seen in Figure 2, the radiotracer demonstrated efficient binding to both PANC02 and U87MG cells in 5 min. After 45 min of incubation, specific binding rates of 31.2 ± 9.4% in PANC02 and 37.7 ± 2.4% in U87MG were observed. There was no statistical difference of binding rate between 30 min and 45 min in both PANC02 and U87MG cells, manifesting the quick completion of preliminary PD-L1 binding.

### 2.3. Biodistribution In Vivo

The PET-CT images showed significantly high accumulation of ^68^Ga-WL12 in PANC02 tumors (Figure 3a, white arrows). Uptake of ^68^Ga-WL12 could be observed in PANC02 tumors as early as 20 min and was retained through 180 min post injection, indicating PD-L1 specificity (Figure 3a). Several representative cross-sectional images and the corresponding sagittal and coronal images at 90 min post injection are presented (Figure 3b). The quantitative analysis of SUV_max_ values of each important organ demonstrated that normal tissues including brain, thyroid, and lung were of very limited uptake; the organs with the highest non-target uptake were the kidneys, liver, and heart, which are involved in blood circulation of the tracer (Figure 3c).

SUV_max_ of tumors, liver, and muscle at different time points was measured to determine a time point with appropriate image parameters. The SUV_max_ of tumors showed a slight downward trend over time, while it peaked at 90 min post injection with 0.38 ± 0.02 (Figure 4a). Tumor-to-liver (T/L) and tumor-to-muscle (T/M) ratios, based on SUV_max_, were calculated. Within 120 min, the highest T/L were obtained at 90 min post injection with 0.45 ± 0.03 (Figure 4b), and T/M exceeded 2 after 90 min (Figure 4c). Based on these observations, the image at 90 min post injection was used in the following section.

### 2.4. PD-L1 Expression

Immunohistochemical assays were performed to detect PD-L1 level of PANC02 xenografts. Hematoxylin and eosin (H&E) staining revealed a good tissue morphology of PANC02 xenografts (Figure 4d). In addition, immunohistochemical staining revealed the moderate PD-L1 expression in the PANC02 xenografts with TPS of 4.6 (Figure 4e).

### 2.5. PET-CT Imaging of Multiple Tracers 

PET-CT images of four commonly used radiotracers were compared in PANC02 xenografts. The images showed significant accumulation of each radiotracer in PANC02 tumors (Figure 5a, white arrows) while with distinct biodistribution in vivo. Further quantitative analysis found that tumor SUV_max_ in ^68^Ga-WL12 was the highest with 0.38 ± 0.02, followed by ^18^F-FDG with 0.32 ± 0.03, and ^68^Ga-RGD showed the lowest SUV_max_ of less than 0.05 (Figure 5b), while ^18^F-FDG showed the highest T/L (Figure 5c) and T/M (Figure 5d) ratio with 2.30 ± 0.49 and 8.22 ± 0.19, respectively, which was significantly higher than the other three radiotracers.

## 3. Discussion

With the rapid development of immune therapy, how to improve response rate of certain cancers and maximize the therapy effect has become the primary issue. PD-L1 antibody have been proven to achieve high response to strong PD-L1-positive cancers. Through further studies, it was found that the level of PD-L1 is related to the curative effect of PD-L1 antibody. Thus, detecting the PD-L1 expression of tumors becomes the key issue. Biopsy and immunohistochemistry are traditional methods to evaluate PD-L1, while these ways are invasive and can only detect a part of lesion locally, limiting their repeat operation during the disease process and exhibiting limitations when there are more than one primary lesion or metastatic lesions. In these cases, non-invasive and dynamic technologies to visualize whole-body PD-L1 are urgently needed. PET can detect target molecule non-invasively in real time, leading to the advent of a series radiotracers labeled with ^68^Ga, ^64^Cu, and ^18^F to target PD-L1. Among these, anti-PD-L1 antibodies are limited as radiotracers due to their long clearance time and the inaccurate imaging being affected by therapeutic anti-PD-L1 antibodies. Small-molecular peptide WL12 has high affinity and can permeate into the tumor within a short time compared with antibodies, showing good clinical feasibility. ^68^Ga-NOTA-WL12 has been proven to show high specificity in PD-L1-positive tumors, and PET imaging demonstrated high tissue contrast preclinically and clinically.

In this study, we evaluated the modified WL12 with HBED-CC as the labeling site and explored its affinity to PD-L1 and distribution in mice pancreatic cancer models. PANC02 cells and U87 cells that are known to have high PD-L1 expression were used to validate the affinity of ^68^Ga-WL12 to PD-L1. ^68^Ga-WL12 demonstrated excellent quality in the first 20 min, and uptake in tumors continued until the end of guided observation (180 min), which has the potential to guide treatments in clinic.

Pancreatic ductal adenocarcinoma (PDAC) is one of the most aggressive cancers and is the third leading cause of cancer death [20]. Thus far, the only potentially curative option for pancreatic cancer remains surgery, but about one-half of patients with PC present with distant metastases, and approximately one-third present with locally advanced diseases [21], for whom it is impossible to perform surgery. Despite introduction of new regimens of chemotherapy and radiotherapy, the overall 5-year survival rate for PC is 8%, and significant improvement in survival over the past decades has remained absent [22]. Thus, the need for novel therapies is warranted. Because of the heterogeneity of PC and complicated tumor microenvironment, some lesions express a low PD-L1 level, and these do not benefit from PD-L1 blockade therapy. However, its combination with other therapies has become a topic of interest because much research has reported the upregulation of PD-L1 after intervention of chemotherapy and radiotherapy [23,24,25,26]. PET imaging of PD-L1 can not only select patients who can response to PD-L1 blockade but also guide the combination regimen by finding the appropriate time point to add PD-L1 blockade. 

In order to detect the tumor from many aspects, at the same time that PD-L1 is evaluated with ^68^Ga-WL12, radiotracers targeting other targets can be performed. ^18^F-FDG is the most commonly used radioactive tracer to evaluate the activity of tumor from a metabolic point of view [27]. The great advantages of FDG-PET for PDAC appears to be the strong correlation between FDG level and tumor aggressiveness, which can predict distant metastasis and survival [28], but it is not effective enough in diagnosing early-stage pancreatic cancer or detecting small metastases. In addition, it is difficult for FDG PET to distinguish between PDAC and pancreatitis because of the same high uptake [29]. Besides glucose metabolism, the uptake of FDG is also closely associated with hypoxia. Hypoxia-inducible factor 1α (HIF-1α) in response to hypoxia not only plays an important role in the uptake of ^18^F-FDG but is also proven to be relevant to the enhancement of PD-L1 expression. Thus, the uptake of FDG at an early phase after immunotherapy could be helpful for therapeutic monitoring [30]. However, in the evaluation of a non-small cell lung cancer patients, the uptake of FDG was intense in the tumor regardless of PD-L1 expression levels, with no significant heterogeneity among or within lesions [18]. Further large-scale prospective studies are warranted.

Fibroblast activation protein (FAPs) are overexpressed by cancer-associated fibroblasts (CAFs) of various cancers, and several radiolabeled FAPI variants have already been introduced as promising targets for PET/CT imaging [31]. Among this, ^68^Ga-FAPI-04 has shown excellent application in clinical diagnosis and superior to ^18^F-FDG in detecting small metastases [32] and distinguishing between PDAC and pancreatitis [33]. CAFs interact with tumor-infiltrating immune cells as well as other immune components within the tumor immune microenvironment (TIME), consequently shaping an immunosuppressive TME that enables cancer cells to evade the surveillance of the immune system. In-depth studies of CAFs and immune microenvironment interactions might provide novel strategies for subsequent targeted immunotherapies [34].

The radiolabeled cyclic arginine-glycine-aspartate (RGD) peptides can bind to integrin αvβ3 and have been regarded as potent molecular agents for imaging angiogenesis and tumors. It has been reported in non-small cell lung cancer (NSCLC) that higher ^18^F-RGD uptake is associated with low PD-L1 expression in tumor cells, and SUV_max_ is the best parameter to display tumoral expression of PD-L1, which indicates that ^18^F-RGD PET may be useful for reflecting the immune status [35]. However, the tumor uptake of RGD-based radiotracers was only moderate. To elevate the tumor uptake, new radiotracers based on RGD have been explored [36]. 

## 4. Materials and Methods

### 4.1. Preparation of Radiopharmaceuticals 

The precursor molecules of DOTA-E-[c(RGDfk)]_2_ and FAPI-04 were obtained from Shanghai Nice-labeling Bio-Technology Co., LTD. HBED-CC-WL-12 was obtained from Shanghai Artery Medical Technology Co., LTD. All of the precursors were of chemical purity above 99%. ^68^Ga was eluted by HCl (0.05 mol/L) from ^68^Ge/^68^Ga-generator (ITG, Garching, Germany). In a typical labeling procedure, 100 µg DOTA-E-[c(RGDfk)]_2_, HBED-CC-WL12, or FAPI-04 dissolved in 1 mL sodium acetate (0.25 mol/L) was mixed with 185 MBq ^68^Ga eluted by 4 mL HCl and reacted at 100 °C for 10 min to obtain ^68^Ga-RGD, ^68^Ga-WL12, or ^68^Ga-FAPI-04, respectively. The products did not require purification post labelling.

### 4.2. Radiochemical Purity and Stability In Vitro

The radiochemical purity of [^68^Ga]-WL12 was determined by thin-layer chromatography (ITLC) using glass microfiber chromatography paper impregnated with a silica gel as the stationary phase and acetonitrile (50%) as the mobile phase. For the evaluation of in vitro stability, [^68^Ga]-WL12 was incubated at 37 °C with saline and 10% serum, respectively, for 60 min. Aliquots (3.7 kBq) of each radiolabeled ligand were taken and analyzed by ITLC.

### 4.3. Cells and Animal Models 

Murine pancreatic cancer cell PANC02 and human glioma cancer cell U87MG were obtained from Tongpai Biotechnology Co., LTD. The cells were cultured in DMEM medium containing 10% fetal bovine serum (FBS) and 1% penicillin–streptomycin. The cells grew adherent when cultured at 37 °C, 5% CO_2_, and appropriate humidity.

C57BL/6 female mice were obtained from Shanghai Lingchang Biotechnology Co., LTD, Shanghai, China. The animals were kept in the Animal Experiment Center of Shanghai Changhai Hospital of SPF level. Then, 1 × 10^6^ PANC02 cells in 0.1 mL mixture of PBS and Matrigel at a 1:1 ratio was inoculated subcutaneous into the right trunk of the mice. When the tumor reached an average volume of 300–500 mm^3^, in vivo experiments of tracer metabolism were performed.

### 4.4. Cellular Uptake Assay

PANC02 and U87MG cells incubated to a confluence of approximately 80% in a 10 cm Petri dish were digested, centrifugated, and separated to several EP tubes. Then, the cells were incubated with 1.5 kBq ^68^Ga-WL12 in a state of constant shaking for 5, 15, 30, and 45 min. At each time point, precipitation of cells was collected after centrifugating and washing by PBS. The radioactivity for each sample was measured by a γ-counter and calculated as the percentage.

### 4.5. PET-CT Imaging

All imaging acquisition was performed on a clinical PET-CT scanner (Biograph64, Siemens Healthcare, Erlangen, Germany). Before imaging, the model mice were injected with 50 μL sodium pentobarbital (3%) into the abdominal cavity for anesthesia. Then, 3.7 MBq tracers were injected intravenously (IV) into PANC02 tumor-bearing mice, in which ^68^Ga-RGD (*n* = 3), ^68^Ga-FAPI-04 (*n* = 3), and ^18^F-FDG (*n* = 3) image scanning was performed at 60 min post injection, while ^68^Ga-WL12 (*n* = 3) image scanning was performed at different time points (20, 40, 60, 90, 120, 150, and 180 min) to observe the distribution in vivo. For quantification of tracer uptake, 3D regions of interest (ROI) were drawn on the muscle, liver, and tumor.

### 4.6. PD-L1 Expression Analysis by IHC Staining

Tumor samples extracted from tumor-bearing mice were fixed with 4% paraformaldehyde. The samples were trimmed, dehydrated, embedded, sliced, stained, and sealed when in good fixation in strict accordance with the instruction of pathological experiment examination. Rabbit monoclonal antibodies against PD-L1 Rabbit anti-human PD-L1 (1:300, Servicebio, Cat. #GB11339A) were used for IHC analysis. Tissue sections were scanned using a panoramic slice scanner (3DHISTECH, PANNORAMIC DESK/MIDI/250/1000). CaseViewer2.4 scanning software was used to select the target area of the slice for imaging. Image-Pro Plus 6.0 analysis software was used to quantify the presence of PD-L1. The level of PD-L1 expression was presented as a tumor proportion score (TPS), which is the percentage of viable tumor cells showing membrane PD-L1 staining relative to all viable tumor cells, and PD-L1 positivity was defined as ≥1% of TPS.

### 4.7. Statistical Analysis

Statistical analyses were performed GraphPad Prism software 8.0. Measurement data were expressed as mean ± standard deviation, and *t*-test was used for comparison between groups. *p* < 0.05 was considered statistically significant.

## 5. Conclusions

We evaluated the HBED-CC-modified WL12 to prove the modified labeling procedure and showed its better biological properties in vitro and in vivo. ^68^Ga-HBED-CC-WL12 can specifically target PD-L1-positive xenografts in a short period of 20 min, which has potential to detect the status of illness and guide immunotherapy.

## Figures and Tables

**Figure 1 pharmaceuticals-16-00080-f001:**
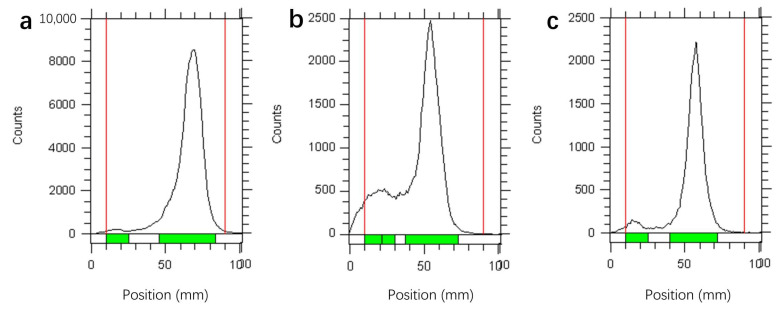
The initial labeling yield of ^68^Ga-WL12 was analyzed by ITLC (**a**), and the stability in vitro was evaluated in FBS (**b**) and PBS (**c**) at 60 min.

**Figure 2 pharmaceuticals-16-00080-f002:**
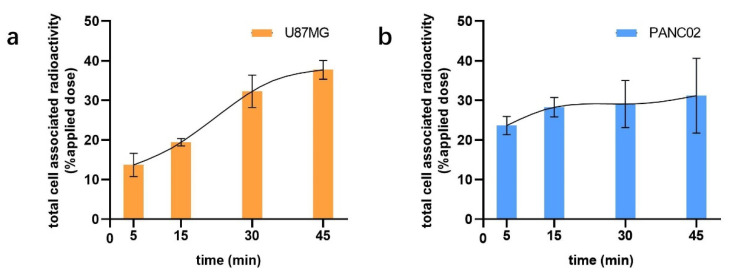
Binding capacity of ^68^Ga-WL12 to U87MG (**a**) and PANC02 (**b**) cells after incubation for 5, 15, 30, and 45 min.

**Figure 3 pharmaceuticals-16-00080-f003:**
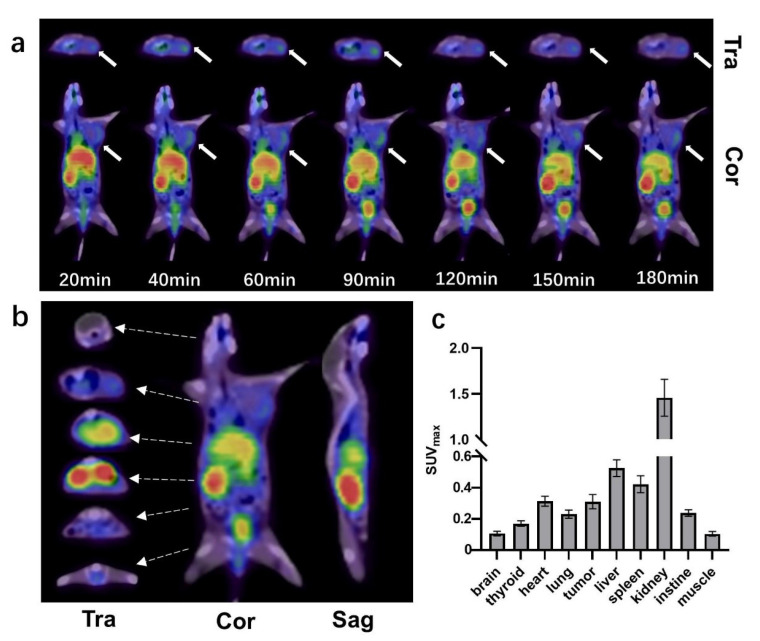
(**a**) PET/CT imaging of ^68^Ga-WL12 in PANC02-bearing mice at 20, 40, 60, 90, 120, 150, and 180 min post injection. Top row, transverse images (Tra); second row, coronal images of tumors (Cor). (**b**) Several representative cross-sectional images and the corresponding sagittal (Sag) and coronal images at 90 min post injection. (**c**) SUV_max_ of critical organs in PANC02-bearing mice (*n* = 3) at 90 min post injection.

**Figure 4 pharmaceuticals-16-00080-f004:**
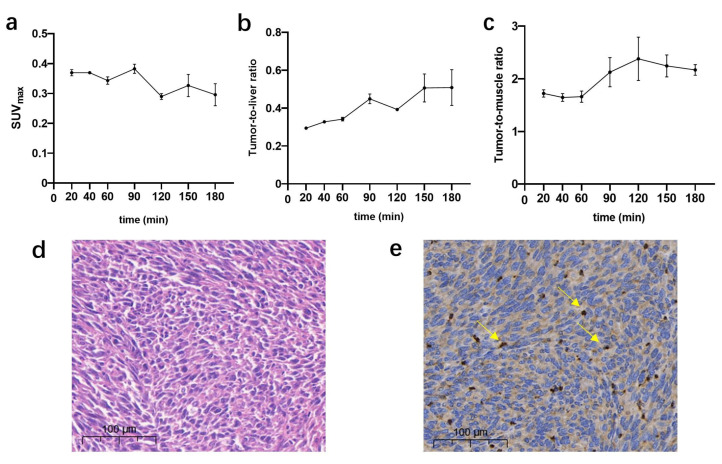
PET-CT uptake data of SUVmax (**a**), tumor-to-liver ratio (T/L) (**b**), and tumor-to-muscle ratio (T/M) (**c**) of xenografts at serial time points. (**d**) Hematoxylin and eosin (H&E) staining of PANC02 tumors (magnification, ×10). (**e**) Immunohistochemical staining of PANC02 tumor xenografts using an PD-L1-alpha antibody (yellow arrows indicate PD-L1-positive cells) (magnification, ×10).

**Figure 5 pharmaceuticals-16-00080-f005:**
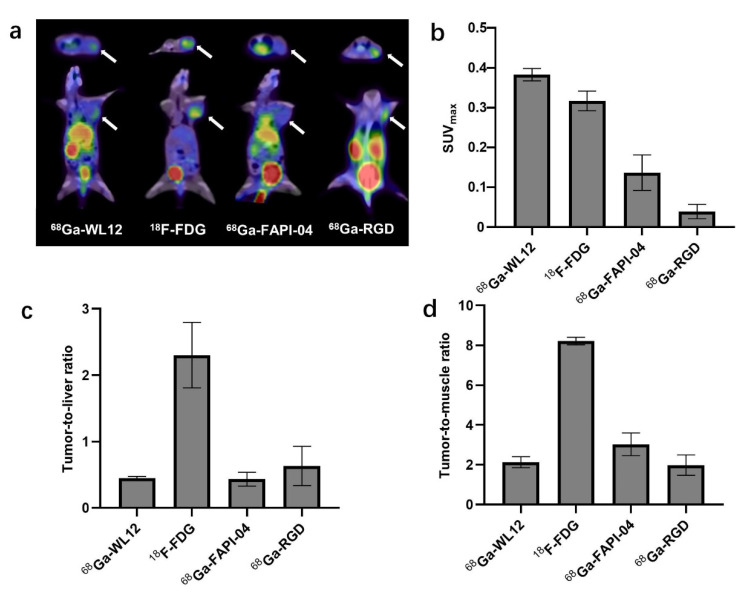
(**a**) PET/CT imaging of ^68^Ga-WL12, ^18^F-FDG, ^68^Ga-FAPI-04, and ^68^Ga-RGD in PANC02-bearing mice. PET-CT uptake data SUV_max_ (**b**), tumor-to-liver ratio (T/L) (**c**), and tumor-to-muscle ratio (T/M) (**d**) of xenografts in these four radiotracers.

## Data Availability

All of the data presented are available in the manuscript.

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
