# Peer review of "^68^Ga-HBED-CC-WL-12 PET in Diagnosing and Differentiating Pancreatic Cancers in Murine Models"

_pharmaceuticals, 2023, doi:10.3390/ph16010080_

Round 1
Reviewer 1 Report
The authors describe the evaluation of a Ga-68 labelled cyclic peptide which targets PD-L1 in a xenograft model of pancreatic cancer. The product shows some favourable features in comparison with alternative agents.
MINOR
Line 2. I think the title could be more informative. In particular, “PanC” should be expanded. Also, it could be made clear that this is preclinical work
Line 77. It would be useful to state the manufacturer of the Ga-68 generator, though the 0.05 M HCl gives a clue
Line 81. Please clarify whether or not the products were purified post labelling. In reference 19, solid phase extraction cartridge purification was used.
Line 138. It is stated that 92.5% RCP meets the requirement for temporary storage. In reference 19, a minimum RCP of 95% was specified.
Lines 143-150. Cellular accumulation studies. Was the number of cells per well or tube standardized between cell lines? Presumably for each cell line the number was the same. Since accumulation was expressed as % of added activity, and no correction was made for cell volume or protein content, the absolute values are not directly comparable, being influenced by total volume, intracellular volume (which can vary between cell lines), cell concentration, and number of cells. Secondly, the authors refer to specific binding. Was non-specific binding assessed, i.e. with excess of cold peptide or non-target-expressing xenografts? A similar statement about specificity is made in the discussion (line 217).
Lines 154-171. PET imaging studies. Since a clinical scanner was used, the authors are working at the lower limit of spatial resolution (compared to a small-animal scanner with much greater resolution), so the delineation of organs and resultant SUV values are somewhat questionable. Ratios of SUVs would be even more variable.
Line 161 states that heat uptake was limited. According to Fig. 3C, heart is actually one of the higher organs
TYPOS ETC
Lines 10-22, Abstract. Superscripts missing for 68Ga and 18F
Line 15. Suggest “showed” rather than “was of”
Line 15. Either remove “can” or change to “can specifically target”
Line 21. Suggest “guiding” rather than “the guiding value for”
Line 28. Second sentence of this paragraph needs rewording. It is not a complete sentence.
Line 31. “tends” rather than “tend”
Line 36. “Research has” rather than “Researches have”
Line 51. This sentence, beginning “There are two…” needs rewording as it is confusing
Line 62. “potential” rather than “potentials”
Line 63. “has” rather than “have”
Line 70. Remove extra space between “Ga” and “-“
Line 107. Suggest removing “used”
Line 163. Do you mean “circulation”? But even that isn’t quite correct. I would suggest deleting the second half of that sencence, from “in other words…”
The references are not consistently in the journal format
The manuscript will require editing for English grammar and idiom
Author Response
Line 2. “PanC” was expanded to “pancreatic cancers”, and “in murine models” makes clear that this is preclinical work.
Line 77.” (ITG, Garching, Germany)” is the manufacturer of the 68Ge/68Ga-generator.
Line 81. The products did not require purification post labelling.
Line 138 The labeling rates we obtained were indeed not as high as the reference 19, which is something we need to improve in the future.
Line 143-150. The number was not the same between the two cell lines because we just focus on observing the uptake pattern of one cell line. Secondly, due to the limited number of precursors obtained, we were unable to perform the experiment with excess of cold peptide or non-target-expressing xenografts. Line 217 refers to the specificity of 68Ga-NOTA-WL12, which has been studied relatively well.
Line 154-171. Although the conditions were limited, the clinical scanner worked well and the presence of multiple experienced teachers delineating the organs greatly reduced the instability.
Line 161. We made the correction that the heart is one of the high uptake organs.
The spelling mistakes have been corrected everywhere. Thank you very much for your advice!
Reviewer 2 Report
PD-L1 expression may be predictive of benefit with immune checkpoint inhibitors. Currently, PD-L1 expression is assessed by immunohistochemistry from tissue samples and is reported as a numerical value (percent positive tumour or immune cells). Therefore, a given result can only represent PD-L1 expression of a small portion of a selected tumour, and there are frequently multiple tumours (e.g., primary, and metastatic sites) in the same patient that are not assessed. A way to start addressing this shortcoming is molecular imaging. Therefore, the topic of this article has a high priority. Moreover, the application of mAbs, as biological vector of the radiopharmaceuticals, limits the use of this technique due to their slow clearance rate and long metabolism, and the selection of a smaller molecule – such as WL12 peptide – seems a logical improvement.
In the description of the experimental work, I have found some inaccuracy:
line 53 – the mentioned slow clearance will result a low target to background ratio, and not high
line 65 – HBED-CC hardly will simplify the labelling procedure. This chelator was successful for PSMA ligand, but for now one can see, that the contribution was not extraordinary. Moreover, for example, NOTA is suitable for room-temperature labelling, and we can see also that authors are using elevated temperature for gallium-incorporation. Therefore, its role rather a “pharmacokinetic modifier”.
line 79 – the preparation of stock solution of peptides in basic media is not good selection. It is better to create an aqueous, concentrated solution from it, and to adjust the pH with the buffer in situ. Using the 4 ml HCl means practically the full eluate. It is partially the explanation of the necessity of the 100 micrograms (which is rather high amount), and if the authors do not want to involve a purity step for the nuclide, a simple fractionated elution will provide them appr. 60% of activity within only 25% of volume.
line 84 - Whatman chromatographic paper is good for paper chromatography and not ITLC. Instant TLC uses a binderless, glass microfiber chromatography paper impregnated with a silica gel
line 134 – with an optimised conditions one can reach a higher RCP with NOTA or HBED than 90%. 3-6 pH is a rather wide range (for example we are using 4-4.5 for NOTA labelling). Figure 1. also indicate that the applied paper chromatography is vulnerable if the decomposition is bigger
chapter 3.3: I have a doubt of the selected cell-lines. Obviously, we can see the accumulation in subcutaneous position, but taking account the background of the liver, spleen and heart and the fact that PANC02 originated from the pancreas, I can not imagine how the authors will be able to distinguish a PANC02 tumour at the original position.
chapter 3.5: Perhaps the colour-code was not identical everywhere, but for me – based on figure 5 (A) – it is not clear the “bad” result of the RGD analogue. Moreover, one can conclude that for the PANC02 tumour the general tumour-seeking 18F-FDG seems better tool than the specific ligands, and – despite the urgent need to evaluate the PD-L1 positivity of tumours, more advanced analogues can be only promising.
Finally a general remark for taking a bigger emphasis on the separation of the words at the end of the rows.
Author Response
Line 53. I'm sorry that the meaning of this sentence is not accurate. What this sentence wants to express is that mAbs takes a long time to get a high target-to-background ratio, while peptide only takes a short time. We corrected “high” to “acceptable”.
Line 65. The word “simplify” is not accurate, because HBED-CC hardly simplify the labelling procedure but optimize the labelling procedure by extending the PH range for which the labeling was applicable. As you suggest, we changed the word “simplify” to “modify” in the text.
Line 79. As a preliminary animal experimental study, the influencing factors such as the volume of elution fluid were not considered. In the actual labeling process, 100 micrograms is not necessary. Based on your comments, we will further optimize the specific activity and concentrated elution for preclinical verification in future.
Line 84. Thank you very much for pointing out this error! After careful verification, we determined that it was indeed glass microfiber chromatography paper impregnated with a silica gel used in the experiment and have made a correction in the text. I'm sorry to have made such a careless mistake.
Line 134. Yes, as you can observe, Figure 1. also indicate that the applied paper chromatography is vulnerable if the decomposition is bigger.
chapter 3.3. We acknowledge the low image resolution of the mouse model, therefore, the subcutaneous tumor model was used. The spatial resolution of the image is good for the patient, we believe that it is not difficult to distinguish the pancreas from the surrounding organs.
chapter 3.5. You are right. The results we have obtained are only partial, further studies on these tracers are needed.
Thank you very much for all your suggestions, which are very helpful to us.